# Single-Domain Antibodies as Antibody–Drug Conjugates: From Promise to Practice—A Systematic Review

**DOI:** 10.3390/cancers16152681

**Published:** 2024-07-27

**Authors:** Víctor Manuel Medina Pérez, Marta Baselga, Alberto J. Schuhmacher

**Affiliations:** 1Molecular Oncology Group, Instituto de Investigación Sanitaria Aragón (IIS Aragón), 50009 Zaragoza, Spain; mbaselga@iisaragon.es; 2Fundación Aragonesa para la Investigación y el Desarrollo (ARAID), 50018 Zaragoza, Spain

**Keywords:** antibody–drug conjugate, ADC, VHH, single-domain antibody conjugates, small format antibody–drug conjugate, single-chain variable fragment, nanobody, sdAb, HCAb

## Abstract

**Simple Summary:**

Antibody–drug conjugates (ADCs) have emerged as a potent cancer therapy by selectively delivering cytotoxic payloads to tumors. However, they face limitations due to acquired resistance and adverse effects. New ADC formats, such as bispecific ADCs and probody–drug conjugates, offer potential solutions. Nevertheless, single-domain antibodies (VHHs), also known as nanobodies, present a promising alternative. VHHs possess unique characteristics over ADCs, including a smaller size, enhanced tissue penetration, and rapid clearance. Their stability, solubility, and manufacturability surpass those of conventional antibodies, enabling cost-effective production and expanding the range of targetable antigens. Therefore, VHHs can mitigate some of the risks associated with conventional ADCs, representing an exciting prospect for next-generation ADCs.

**Abstract:**

Background: Antibody–drug conjugates (ADCs) represent potent cancer therapies that deliver highly toxic drugs to tumor cells precisely, thus allowing for targeted treatment and significantly reducing off-target effects. Despite their effectiveness, ADCs can face limitations due to acquired resistance and potential side effects. Objectives: This study focuses on advances in various ADC components to improve both the efficacy and safety of these agents, and includes the analysis of several novel ADC formats. This work assesses whether the unique features of VHHs—such as their small size, enhanced tissue penetration, stability, and cost-effectiveness—make them a viable alternative to conventional antibodies for ADCs and reviews their current status in ADC development. Methods: Following PRISMA guidelines, this study focused on VHHs as components of ADCs, examining advancements and prospects from 1 January 2014 to 30 June 2024. Searches were conducted in PubMed, Cochrane Library, ScienceDirect and LILACS using specific terms related to ADCs and single-domain antibodies. Retrieved articles were rigorously evaluated, excluding duplicates and non-qualifying studies. The selected peer-reviewed articles were analyzed for quality and synthesized to highlight advancements, methods, payloads, and future directions in ADC research. Results: VHHs offer significant advantages for drug conjugation over conventional antibodies due to their smaller size and structure, which enhance tissue penetration and enable access to previously inaccessible epitopes. Their superior stability, solubility, and manufacturability facilitate cost-effective production and expand the range of targetable antigens. Additionally, some VHHs can naturally cross the blood–brain barrier or be easily modified to favor their penetration, making them promising for targeting brain tumors and metastases. Although no VHH–drug conjugates (nADC or nanoADC) are currently in the clinical arena, preclinical studies have explored various conjugation methods and linkers. Conclusions: While ADCs are transforming cancer treatment, their unique mechanisms and associated toxicities challenge traditional views on bioavailability and vary with different tumor types. Severe toxicities, often linked to compound instability, off-target effects, and nonspecific blood cell interactions, highlight the need for better understanding. Conversely, the rapid distribution, tumor penetration, and clearance of VHHs could be advantageous, potentially reducing toxicity by minimizing prolonged exposure. These attributes make single-domain antibodies strong candidates for the next generation of ADCs, potentially enhancing both efficacy and safety.

## 1. Introduction

For decades, chemotherapy based on cytotoxic agents has been the predominant treatment for a broad spectrum of cancers and continues to be the most commonly prescribed therapy in oncology [1]. Traditional chemotherapy presents several limitations, including low selectivity, limited curative efficacy, easy development of drug resistance, and significant side effects. In the early 20th century, Paul Ehrlich introduced the concept of “magic bullets”, proposing that specific compounds could directly target diseased cells [2,3]. Monoclonal antibodies (mAbs) were a pivotal development in realizing this concept, as they can precisely detect antigens on the surface of tumor cells. The therapeutic use of mAbs began after Georges Kohler and César Milstein introduced the hybridoma technique for producing mAbs in 1975. The first mAb, clinically approved in 1986, was Muromonab-CD3 (Orthoclone OKT3, a murine-derived mAb targeting CD3) [4,5]. To date, more than 100 mAbs have been approved by the US Food and Drug Administration (FDA) for the treatment of various human diseases, including cancer, autoimmune diseases, and chronic inflammatory diseases [1,3,6,7,8,9].

Antibodies have antitumor action by being antagonists or agonists of their receptors in the cell or by sequestering the ligands of signaling pathways such as VEGF. In addition, they trigger the action of complement or the recruitment of immune cells. On their own, they have limited activity, and complete responses in cancer are rarely recorded with the use of mAbs alone [4,5,10].

Although mAbs have made significant strides in disease therapeutics, their standalone use is often insufficient, likely due to their lower effectiveness against cancer cells compared to chemotherapy. By attaching a highly cytotoxic small molecule to a mAb, the antitumor efficacy can be greatly enhanced, creating a new type of antibody derivative called an antibody–drug conjugate (ADC). These ADCs can selectively deliver potent small-molecule drugs directly to targeted cancer cells, inducing apoptosis and effectively serving as “magic bullets” [7,11,12]. The development of genetic engineering, DNA editing techniques, and forms of antibody production have facilitated the new generation of ADC against cancer [2,3,7]. ADCs are a three-component construct comprising the antibody, the binding linker, and the payload. The target specificity of the mAb allows for theoretical selectivity for the tumor cell without damaging non-target cells lacking the antigen, thus improving safety and increasing the therapeutic window [2,7].

In 1989, Raymond Hamers, Cécile Casterman and Serge Muyldermans at the Vrije Universiteit Brussel (Belgium) discovered a new type of antibody—serendipitously—while analyzing total and fractionated immunoglobulin G (IgG) molecules in the serum of a dromedary. This camelid antibody was smaller and simpler than the conventional ones found in mice and humans, composed of two heavy chains and two light chains. The described smaller IgG subclasses appeared to lack light chains. Published in 1993, this research demonstrated that the structure in camels is different, with up to 75% of the immunoglobulins in the plasma of dromedaries having a much smaller molecular weight and containing only one constant chain, termed heavy chain-only antibodies (HCAbs) [13]. Similar structures were later described in sharks, known as shark immunoglobulin new antigen receptors (IgNARs) [14].

The study of these structures and the subsequent recognition that the variable domains of these antibodies (VHHs or Nanobodies^®^ and VNARs, respectively) function autonomously as single-domain antibodies (VHHs) has opened up an impressive field of biotechnological development based on them [15,16]. Since the FDA approved the first VHH antibody in 2019, VHHs have become an alternative to traditional antibodies due to their potential advantage of unique structure and stability [17,18]. To date, four VHHs have received approval for the treatment of multiple diseases, and clinical trials are underway for various medical applications, mainly for imaging and chimeric antigen receptor (CAR) therapies [19,20].

This systematic review aims to assess advances and future prospects in using VHHs as ADC and identifies possible theoretical improvements and new emerging therapies in the field.

## 2. Materials and Methods

### 2.1. Design of the Study and Methods Used for Finding Information

This review was conducted in accordance with the Preferred Reporting Items for Systematic Reviews and Meta-Analyses (PRISMA) guideline [21]. The study was designed to conduct a thorough review of current research on VHHs and their applications as a class of ADCs. The review encompassed a range of relevant findings related to the topic and provided a comprehensive understanding of the current state of research in this field. The main advances and theoretical ways to enhance the results are discussed, along with future perspectives in the field of conjugated VHHs. This Systematic Review has not registered in a public registry.

### 2.2. Strategy for Finding Information

The structured inquiry included pertinent research studies released from 1 January 2014 to 30 June 2024). This specific period was chosen to capture the most recent and relevant developments in the field of ADCs and VHHs. Although the first ADC was approved by the FDA in 2000 (gemtuzumab ozogamicin), the field has significantly evolved since then. Additionally, the first single-domain antibody (caplacizumab) was FDA-approved in 2019. By selecting a period starting from five years prior to the approval of caplacizumab, we aimed to include contemporary studies and advancements that reflect the current state of research and therapeutic applications.

The databases used for the search were PubMed, Cochrane Library, ScienceDirect and LILACS. The search criteria included a mix of medical subject headings (MeSH) and keywords related to ADC and nanoantibodies, such as “nanobody conjugates”, “single-domain antibody conjugates”, “drug-bound VHHs”, “small format antibody-drug conjugate”, “scFv”, “single-chain variable fragment”, “single-domain antibody-drug conjugates”, “fully human single-domain antibody-drug conjugates”, “sdAb”, “NDC”, “small ADC”, “functional heavy (H)-chain antibody”, and “HCAb”. The terms were combined effectively for the search strategy using the Boolean operators “AND” and “OR”. Articles that were published in either English or Spanish and can be fully accessed online were considered. Research covered both clinical and preclinical studies. 

### 2.3. Evaluation and Selection

The first search produced a detailed collection of articles, which were later refined using automated tools to exclude irrelevant records according to the inclusion criteria (Figure 1). After eliminating duplicates, three authors (VMMP, MB, AJS) individually reviewed the titles, abstracts, and full texts of the remaining articles. Additionally, other relevant studies selected by the team outside the specified time range were also incorporated. Full-text reviews were carried out for articles that met the inclusion criteria, and any discrepancies were resolved through consensus-based discussions.

### 2.4. Determination of Quality

The articles were included in the final analysis after evaluating their quality of evidence. Only peer-reviewed journal articles with rigorous methodologies were reviewed, such as randomized controlled trials and cohort studies, to ensure high-quality evidence synthesis. Case reports, editor letters, and conference studies were excluded, because they typically do not provide the level of rigor and comprehensive data needed for a systematic review. The chosen articles examined different facets of advancements in target VHHs, conjugation methods, payloads, and future prospects in the field of ADCs.

### 2.5. Extraction and Analysis of Data

Information from the studies was collected and combined to offer a thorough overview of the present research status. The results were considered in relation to the key developments, theoretical enhancements, and upcoming research areas in ADC and VHH-ADCs. The outcomes were summarized in a PRISMA flow diagram to demonstrate the selection process and results (Figure 1). 

The review focuses on various aspects of ADCs, including the evolution of the targets, the linker or forms of conjugation methods used, the payload, the current research in the field, VHHs as ADCs, and future perspectives.

## 3. ADC: A Look at Its Evolution

An ADC consists of a mAb linked to a potent cytotoxic payload via a chemical linker. This molecular design merges the target specificity and extended circulation half-life of an antibody with the high cytotoxic potency of antitumor agents that are too toxic for standalone use. Consequently, compared with conventional chemotherapies, ADCs can offer enhanced antitumor efficacy, leading to improved clinical benefits and quality of life outcomes [1,22,23]. ADCs offer several advantages due to the precise binding of the antibody and the pro-apoptotic nature of the payload. These include high therapeutic efficacy, high specificity, relatively lower toxicity to non-cancer cells and reduced side effects compared to conventional chemical cancer treatments [6,22,23,24,25,26,27,28,29,30].

To build an ADC, three main components have to be considered: a mAb, a cytotoxic payload, and a linker (Figure 2). The mAb (typically IgG) facilitates the internalization of the ADC into target cells via receptor-mediated endocytosis, allows for a prolonged presence in the bloodstream and reduces the likelihood of triggering an immune response [1,24,25]. The payload is a potent cytotoxic agent intended to kill cancer cells (e.g., microtubule, DNA synthesis and topoisomerase inhibitors) [1,24,25]. The ratio of drug molecules to antibody molecules is optimized for effectiveness and safety. The linker connects the mAb to the toxic payload [1,24,25]. When an ADC binds to a target antigen on tumor cells, it can deliver a cytotoxic payload directly into the cytoplasm of the targeted cell through receptor-mediated endocytosis. Alternatively, the cytotoxic drug may be released during lysosomal degradation, disrupting DNA or inhibiting cell division, ultimately killing tumor cells. Effective drug targeting, which ensures high tumor specificity and efficient internalization by cancer cells, is a critical determinant of an ADC’s druggability [1,24,25]. The ideal characteristics of ADCs include [1,2,6,7,12,22,23,24,25,26,27,28,29,30,31]:The antibody composing an ADC should present high stability and high affinity for the target and deep tumor penetration. Low/no immunogenic potential.The target should be a surface-exposed (or extracellular) antigen, serving as the delivery address.The linkers should be stable before reaching the targeted tumor site.ADC should be efficiently internalized via any of the endocytosis pathways and successfully trafficked to lysosomes, where they accumulate.Payloads from ADCs should be rapidly released upon entry into lysosomes.The linked drug should be capable of efficient cell killing. Usually, payloads present higher toxicity than other chemotherapeutic agents (from 100- to 1000-fold). Importantly, the potency of the cytotoxic payload should be directed by conjugating it to a tumor-specific antibody.

**Figure 2 cancers-16-02681-f002:**
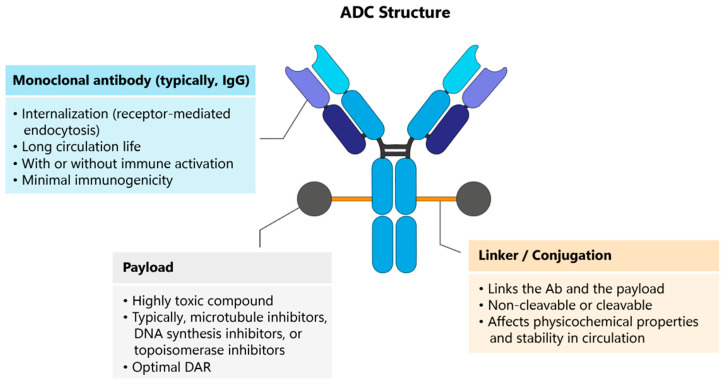
Schematic representation of the ADC structure. The generic structure of the ADC consists of three parts: the antibody, the payload and the linker between the payload and the antibody.

Each component can influence the final efficacy and safety of an ADC. ADC development must carefully consider each of these elements, including the selection of the target antigen, antibody, cytotoxic payload, linker, and conjugation methods [1,22,23,24,25,26,27]. Current ADCs have advanced in engineering the antibody, predominantly using conventional immunoglobulin G (IgG) antibodies, which possess a high affinity that enables effective internalization while preserving plasma half-life. Improvements have also been made in the linker, conjugation site, small-molecule payload, and the average number of drug molecules linked to each antibody or drug-to-antibody ratio (DAR). However, the relationship between the heterogeneous intratumoral distribution and the efficacy of ADCs is still poorly understood [6,22,23,24,25,26,27,28,29,30,31].

### 3.1. From Antibody to ADC

#### 3.1.1. Linkers and Conjugation Process

The chemical linker that connects the antibody to the cytotoxic payload plays a crucial role in ADC design. Various factors, such as the chemistry of the linker, the conjugation strategy, and the site of conjugation, critically influence the pharmacokinetic and pharmacodynamic properties of the ADC. Several approaches can be used for coupling the linker to the antibody, including site-directed linkage to specific amino acids—which is the most commonly used in clinical practice and approved ADCs—and random conjugation to lysine or cysteine residues. Lysine linkage is based on the coupling of an amine from the mAb and an activated carboxylic acid of the payload, while the cysteine-based coupling of mAb reacts cysteine residues with a thiol-reactive functional group in the payload. These two methods generate random conjugates with different DAR and low homogeneity, since the payloads can couple on numerous residues along the mAb. However, other approaches exist to incorporate non-natural amino acids through genetic engineering into the antibody peptidic sequence with specific residual groups that favor site-directed conjugation [32,33]. The coupling of the payload to the mAb can also be generated by enzymatic strategies leading to tightly controlled DARs, as they show high site-specificity. The main enzyme-mediated strategies are transpeptidation using sortase A, microbial transglutaminase or N-Glycan engineering [34,35,36]. Protein conjugates, particularly in the production of bispecific antibodies, represent a rapidly advancing field of interest, with evolving technologies poised to advance conjugation techniques. This advancement enables the creation of site-specific ADCs without necessitating extensive antibody engineering. Recent applications have demonstrated the successful generation of potent bispecific antibodies, underscoring the versatility of chemical methods and their potential in advancing targeted therapeutic development. Specific site conjugation has significantly enhanced ADC development by allowing for precise control over drug linkage sites, thus improving product uniformity and therapeutic profile. While cysteine conjugation currently dominates among clinically approved ADCs, emerging strategies like AJICAP and AJICAP-M hold promise to further enhance the quality and versatility of ADCs [37]. The AJICAP and AJICAP-M technologies are prominent examples of advanced conjugation techniques. AJICAP utilizes maleimide-based chemistry to conjugate drugs to specific cysteine residues on antibodies. AJICAP-M expands this technology to accommodate a broader range of drugs and antibodies with varying specificities and affinities [38]. These conjugation methods offer significant advantages, including the enhanced reproducibility and uniformity of conjugates, reduced drug quantities required for conjugation with process optimization, improved purification, and a minimized risk of free drug in the final product [38,39]. AJICAP-M enables the conjugation of a wider array of drugs using the same technique, thereby facilitating payload diversification without significant procedural changes [38,39,40].

Linkers must maintain the ADC’s stability in the bloodstream to ensure it reaches the cancer cell intact, but they must also be able to be easily cleaved upon internalization (cleavable linkers) to release the payload. Alternatively, non-cleavable linkers should form an innocuous element after releasing the payload, allowing it to still exert its therapeutic effect [6,22,23,25,26,27,28,29,30,31]. Cleavable linkers are designed to be processed at the tumor site, taking advantage of the unique properties of the tumor microenvironment over healthy tissues or systemic circulation. They can be released from the antibody by changes in pH or by enzymes present in the intercellular and intracellular space. These types of linkers generate a membrane-permeable neutral payload capable of promoting bystander killing [31,41,42,43,44,45]. The decision to select one of the two types of linkers hinges on the anticipated therapeutic outcome and the specifics of the targeted tumor. Approximately two-thirds of ADCs in clinical trials use cleavable linkers. The most frequently used are dipeptide, disulfide, and enzyme-cleavable, particularly hydrazone, cathepsin B-responsive disulfide and pyrophosphate diester linkers [34,35,36]. Non-cleavable linkers require the degradation of the antibody component before the drug is released. This mechanism may reduce the bystander effect. However, the advantages of non-cleavable linkers include enhanced stability and a reduced risk of unintended side effects [41,42,43]. The use of cleavable or non-cleavable linkers hinges on the anticipated therapeutic outcome and the targeted tumor. 

#### 3.1.2. Payload

Conventional chemotherapeutic drugs, when used as ADCs, often fail to eliminate malignant cells due to the low DAR and the limited number of antibody-targeting receptors on a tumor cell’s surface [46]. Radioisotope biodistribution and intratumoral concentration studies have shown that only approximately 1–2% of the administered dose of an ADC reaches the tumor [3,22,47,48]. These data underscore the necessity for the chemotherapeutic agent carried by the antibody to be potent enough to induce tumor cell killing at very low concentrations. The payloads used for ADCs are typically far more toxic than conventional chemotherapies, exhibiting sub-nanomolar or even picomolar cytotoxicity in vitro, compared to the micromolar range of several common chemotherapies [47,48,49,50,51,52]. The ideal cytotoxic payload for targeted cancer therapy should present low molecular weight, suitable solubility in water-based buffers, stability in the acidic lysosomal environment, and the ability to retain cytotoxicity even after degradation into linker residue-payload form. They should also exhibit low immunogenicity, since they play a crucial role in the tumor immune microenvironment and can influence the immune response [50,51,52,53,54,55].

Compared to the nanomolar IC_50_ (half-maximal inhibitory concentration) values typical of microtubule inhibitors, some DNA-damaging agents exhibit IC_50_ values in the picomolar range. As a result, ADCs conjugated with DNA-damaging agents can sometimes be more effective. They may operate independently of the cell cycle—unlike tubulin inhibitors, which primarily act during mitosis—and can be effective even in cells with a low expression of the targeted antigen [51,54,55,56,57].

The incorporation of non-traditional payloads into ADCs represents a revolutionary development in the treatment of cancer. Conventional chemotherapy medicines frequently have a limited therapeutic window, which results in substantial toxicities that reduce their effectiveness. On the other hand, ADCs with non-traditional payloads use the accuracy of mAbs to target cancer cells specifically, sparing healthy organs and minimizing systemic adverse effects. The novel conjugations appear to be novel ADCs, which may be categorized as follows:Radioimmunoconjugates (RICS): Over the past ten years, there has been a substantial advancement in the conjugation of radioisotopes for both diagnosis and treatment. This therapy directs irradiation from radionuclides to tumor targets by using mAbs that bind to tumor antigens. The acceptance of Actinium and Lutetium conjugates has cleared the path for numerous therapeutic pairings involving mAbs or VHHs, exhibiting effectiveness in situations unresponsive to prior interventions. Notably, preclinical research on several VHH antibodies, including those that target PDL1 and HER2, has produced encouraging results. Additionally, by utilizing the human IgG1 Fc domain to increase the serum half-life of a CAIX-VHH enzyme-inhibiting antibody, researchers have created constructs that can be labeled with [^89^Zr]Zr(IV) for preclinical PET/CT imaging in mice suffering from colon cancer. Furthermore, [^89^Zr]Zr has proven to be an excellent radiolabeler for anti-CLDN18.2 VHH-ABD and anti-CLDN18.2 VHH-Fc, enabling noninvasive imaging and the quantification of CLDN18.2 expression in gastric cancer. These developments demonstrate how radioisotope conjugation can improve targeted cancer treatments and theragnostics [58,59,60,61,62,63,64,65,66].Immune-stimulating antibody conjugates (ISACs): In the process of developing new cancer drugs, immunological adjuvant compounds that interact with pattern-recognition receptors (PRRs) have entered the stage. An innovative approach to activating localized innate immunity involves the systemic administration of antibodies linked with specific PRR agonists. ISACs have been shown to have potential benefits over traditional ADCs that contain cytotoxic payloads, according to preclinical assessments. Particularly promising are ISACs that used Toll-Like Receptor agonist payloads (TLR7, TLR8, and TLR9) and Stimulators of Interferon Genes (STING) agonist payloads. In order to learn more about the effectiveness of these cutting-edge treatments, a phase I/II clinical trial (NCT05954143) is presently recruiting patients with advanced HER2-expressing solid tumors [62,63,67].Antibody-based protein degraders (degradation-activating compounds or DACs): Agonists and targeted protein degraders (TPDs), using proteolytic targeting chimera (PROTAC) and other molecular glue degraders-based strategies, have attracted considerable attention in current research. In addition to the DACs designed to degrade specific cytosolic proteins, several labs have advanced methods to degrade cell surface proteins using antibody-based approaches. These include antibody-based PROTACs (AbTACs), which use antibodies as carriers to drive targeted protein degradation, proteolysis-targeted antibodies (PROTABs), and lysosome-targeted chimeras (LYTACs) [68,69].Dual-drug ADC or bispecific drug conjugates and other constructs: Bispecific antibody–drug conjugates (bsADC) combine the advantages of ADCs and bispecific antibodies. Dual-specific targeting has the potential to improve the efficacy and safety of ADCs by improving their specificity, affinity, and internalization potential. Preclinical studies have shown that the bispecific ADC concept could lead to the development of more effective anticancer therapies than monospecific ADCs. One study reported that co-administering a HER2 × prolactin receptor (PRLR) bispecific antibody (bsAb) with an anti-HER2 ADC significantly enhanced the cytotoxic activity of the ADC, with the bsADC HER2 × PRLR showing approximately a 100-fold reduction in IC_50_ against the T47D/HER2 cell line compared to the anti-HER2 ADC (0.4 nM vs. 40 nM, respectively) [70]. Other strategies under investigation include a novel bsADC targeting HER2 and HER3, which has shown high therapeutic efficacy in treating breast cancer. These advances underscore the potential of bsADC to advance precision cancer treatments [70,71,72,73,74,75,76,77].

The quantity of drug molecules bound to the antibody, known as the DAR, their hydrophobicity, intertumoral sensitivity, and whether or not they are substrates for multidrug resistance (MDR)-like flow molecules are also important factors when selecting which payload to use [78,79,80,81].

#### 3.1.3. Mechanism of Action of Conjugated Antibodies

Pharmacologically, the action of an ADC can be outlined in four steps: systemic circulation, the Enhanced Permeability and Retention (EPR) effect including passive targeting, penetration within the tumor tissue, and action on cells, which encompasses active targeting and controlled release. The canonical model for the mechanism of action of ADCs can be divided into several stages: the binding of the mAb to the target antigen, the internalization of the molecule, and finally, the cleavage of the linker with the release of the cytostatic payload [50,79,80,81] (Figure 3a–c). After antigen binding on the cell surface, the ADC is internalized by the tumor cell and endocytosed to form an early endosome. Here, the mAb binds to the targeted antigens uniquely expressed in cancer cells via one of three main pathways: clathrin-mediated endocytosis, caveolae-mediated endocytosis, or pinocytosis. The latter is antigen-independent, whereas the first two are antigen-dependent [6,80,81,82,83]. The antigen–ADC complex is then internalized via receptor-mediated endocytosis and trafficked into the lysosome (Figure 3b). Here, the ADC is processed according to the physicochemical properties of the linker, and releases the cytotoxic warhead. Once in the cytoplasm, the released drug ultimately triggers cell death or apoptosis, generally targeting DNA or tubulin. Lipophilic drugs can diffuse from ADC target cells to neighboring cells, killing them independently of their target expression, a mechanism known as the bystander effect (Figure 3d). The antibody can either activate or inhibit target receptors. Designed to specifically recognize and bind to antigens on certain cells, such as tumor cells, ADCs can exert agonistic or antagonistic effects upon binding to their cell surface receptors. This interaction can alter intracellular pathways, potentially inhibiting cell growth or metabolism depending on the design and cellular context (Figure 3e). This effect not only increases the cytotoxicity of ADCs, but also makes it possible to target tumors with heterogeneous antigen expression, increasing the patient population that could benefit [6,22,29,81,82,83,84,85,86]. In addition to the payload-induced killing mechanisms, classical antibody functions, such as the inhibition of the downstream signaling pathways of the target receptor through the Fab region, or Fc-mediated killing mechanisms, such as antibody-dependent cellular cytotoxicity (ADCC) (Figure 3f) and complement-dependent cytotoxicity (CDC) (Figure 3g), as well as antibody-dependent phagocytosis (ADCP) (Figure 3h), directly involve innate or complementary immune effectors [6,56,85,87,88].

However, while these Fc-mediated processes potentially enhance the ADCs’ antitumor effect, they may also adversely affect their safety profile by increasing healthy tissue exposure through nonspecific drug diffusion, Fc-mediated uptake by immune cells, or recycling via neonatal Fc receptors (FcRn) [89,90,91,92,93,94,95,96,97,98,99,100]. Additional factors influencing antibody clearance include the mononuclear phagocyte system and FcRn-mediated recycling. FcRn binds to ADCs within the endocytic vacuole and facilitates their export to the extracellular compartment for recycling [89,90,91,92,93,94,95,96,97,98,99,100].

### 3.2. ADCs Evolution

#### 3.2.1. First-Generation ADCs

In the early development of ADCs, murine antibodies linked to standard chemotherapeutic agents were used, including mitomycin C, N-acetylmarflan, and anthracyclines via stable, non-cleavable linkers such as succinimide or amide spacers. However, this approach led to these antibodies being recognized as foreign by the human immune system, which in turn produced human anti-mouse antibodies, resulting in the rapid elimination of these ADCs from the human body. Moreover, the linkers were not sufficiently stable in the bloodstream, contributing to the ADCs having a brief duration of action. Additionally, the cytotoxic agents sometimes failed to achieve the required levels of efficacy due to their insufficient toxicity in place at the administered dosages [101,102,103]. This early phase was exemplified by the launch of the first human clinical trial for an ADC in 1983, which involved a conjugate of an anti-carcinoembryonic antigen–antibody conjugated with vindesine. Administered to patients with various types of advanced metastatic carcinomas, this ADC was deemed safe and showed signs of effectiveness. However, the initial challenges led to advancements in ADC technology, including the transition from murine to humanized antibodies, enhancing their compatibility with the human immune system and improving the overall therapeutic potential of ADCs [35,101,102,103,104]. Among the most significant weaknesses that limited the efficacy of these initial ADCs were the low potency of the chemotherapeutic agent, the instability of the binding of the drug to the antibody, and low antigen selectivity [35,103,104,105].

#### 3.2.2. Second-Generation ADCs

Second-generation ADCs feature improved characteristics of chemotherapy drugs, being significantly more potent (100- to 1000-fold) and consisting of smaller molecules compared to first-generation ADCs. The use of stronger cytotoxic payloads, such as maytansinoids and auristatins, which offer enhanced binding capabilities and improved water solubility, represent a major advancement in the development of second-generation ADCs. Additionally, modifications to linkers have enhanced plasma stability and ensured a uniform DAR, further improving the therapeutic efficacy and safety of these advanced therapeutic agents [6,35,103,105].

#### 3.2.3. Third-Generation ADCs

Third-generation ADCs represent the most advanced constructs to date, featuring improvements across all three components. They use more specific antibodies that are humanized to reduce immunogenicity. The cargo includes more potent payloads. Additionally, these ADCs incorporate more stable linkers connected through more complex chemistry. This advancement results in stable and reproducible DAR, enhancing their stability in circulation and providing better therapeutic windows [6,26,35,103,104,105,106,107,108].

#### 3.2.4. The Next Generations of ADCs

The next generations of ADCs must strive to overcome persistent limitations, particularly in improving tissue penetration and reducing off-target effects. One strategy to achieve these objectives involves reducing the size of the antibody. VHHs are an antibody format with exceptional characteristics that make them highly promising for ADC development (Table 1). The recent advances in this area are reviewed in the following sections [17,26,63,71,109,110].

### 3.3. Disadvantages of Conventional Antibodies for ADCs

mAb are complex macromolecules, frequently employed as targeting moieties, that face several challenges. Factors such as susceptibility to misfolding in the variable region and high host immunogenicity significantly hinder the development and application of antibody-based therapies. Furthermore, the large size of mAbs hinders their ability to extravasate and effectively penetrate tissues to reach all target cells. Often, poor efficacy arises from the non-uniform distribution of the mAb-based agent within the tumor. Larger molecules diffuse much more slowly compared to the pressure-driven advective transport [111,112,113]. 

Currently, there is no single class of antibodies that possesses all the desired properties required for effective targeting agents in ADCs, including low immunogenicity, rapid distribution, the quick clearance of unbound molecules, and high accumulation in tumors [114,115,116].

## 4. VHHs as Nano-ADCs

HCAbs were found in *Camelidae* (Bactrian and camels, alpacas, and llamas), as well as cartilaginous fish (e.g., sharks, rays, and skates). The camelid-derived VHHs are a unique, functional single-domain of HCAb (Figure 4a). The variable domain of the heavy chain in an HCAb retains high antigen-binding affinity despite presenting one-tenth of the MW of a conventional IgG (12–17 kDa vs. ~150 kDa), being the smallest naturally derived Ag-binding fragment [114,115,116].

### 4.1. VHH’s Physical, Chemical and Structural Properties

The crystal structure of the VHH domain revealed dimensions of 4 nm × 2.5 nm × 3 nm. VHH domains have been found to be highly soluble and more stable than conventional antibodies. They can be stored at 4 °C or −20 °C for months without significantly losing their antigen-binding capacity. Furthermore, the homology between the VHH and VH domains of the human Ig family VH III was found to be greater than 80%, suggesting that the VHH sequence may induce a mild immunogenic response when used in cancer immunotherapy [111,116].

VHH domains can endure harsh conditions, such as a wide pH range (3–9) and extreme chemical (e.g., 6–8 M urea concentration) and thermal denaturing conditions (e.g., maintaining antigen-binding activity after prolonged incubation at 90 °C). This robustness allows for various administration routes, such as intravenous, oral or intraperitoneal [117,118]. Notably, VHH domains possess a fully hydrophilic surface, enhancing their stability and solubility compared to IgG VH domains, and exhibit significantly less aggregation during production or multimerization (e.g., tandem VHH-based multispecific antibodies). The CDR3 loop in camelid VHH domains is typically longer (3–28 amino acids) than in the conventional VH domains of human IgG (8–15 amino acids). This extended CDR3, which determines recognition specificity, increases the potential interaction surface with a target antigen in the absence of a VL domain (Figure 4b). Interestingly, the longer CDR3 in VHH domains can form a finger-like appendage that fits into a protein cleft, enabling the recognition of epitopes that are inaccessible to larger antibodies such as mAbs. However, the small size of the VHH domain results in rapid renal clearance (half-life ~2 h), which is a significant disadvantage for their application in cancer treatment [111,112,113,114,115,116,119,120]. On the other hand, the VHH fragment of the HCAb in serum shows a unique thermo-reversible stability profile. VHHs can withstand high temperatures due to their ability to refold after heat denaturation [121,122,123,124,125,126]. Furthermore, VHHs remain stable under extreme pH conditions, preserving their bioactivity in the stomach or intestine. This stability allows for the design of treatments using various administration methods, including intravenous injection, inhalation, and oral and intranasal delivery. Overall, the stable biochemical and biophysical properties of VHH support their expanding applications in various therapeutic areas [20,121,127,128,129,130].

### 4.2. VHH’s Biological Functions

VHHs have exceptional properties, including their small size and ability to penetrate tumors in vivo, which enhance their tumor-targeting capabilities. Unlike conventional antibodies, VHHs retain the same antigen-binding characteristics as a single immunoglobulin variable domain for antigen recognition [120]. Furthermore, VHHs can access epitopes that conventional antibodies cannot, such as clefts on a protein’s surface [131]. Uniquely, the strict monomeric state of VHHs facilitates independent antigen recognition and binding. Another critical difference between the VHH domain of camelid antibodies and the VH domain of conventional antibodies is found in the FR2 fragment. X-ray crystallography analysis of the VHH protein–antigen complex has shown that hydrophobic amino acids in the VHH FR2 are replaced by hydrophilic amino acids in the FR2 of conventional antibodies. Specifically, the amino acids Phe-42, Glu-49, Arg-50, and Gly-52 in the conventional VH-VL cross-linking FR2 are substituted with Val, Gly, Leu, and Trp, respectively [113,132,133,134,135,136,137]. Additionally, the single-variable domains of VHHs can readily form concave surfaces that function as active sites or receptor-binding pockets. While the extension of the HV loop increases flexibility, it can compromise the stability of the VHH domain’s internal structure. To counteract the flexibility introduced by the longer H3 loop, an extra interloop disulfide bond between the H1 and H3 loops reinforces the extended HV loop structure, thereby enhancing antigen binding [138,139]. Additionally, the CDR3 region in VHHs can form an exposed ring structure, acting like a ‘finger’ that inserts into an antigen’s ‘pocket’, unlike conventional antibodies that usually interact with flat surfaces. An extra cysteine residue in CDR3 can also form a disulfide bond with an additional cysteine residue in either CDR1 or the framework region 2 (FR2), which enhances the stability of the VHH structure and lowers the energy required for antigen binding [140,141].

The rate of passive diffusion of a molecule in tissue is inversely proportional to its molecular size. Consequently, monovalent VHHs (12–17 kDa) exhibit faster vascular permeability and better tissue penetration compared to conventional antibodies (~150 kDa) enabling VHHs to achieve a more homogeneous distribution, such as in solid tumors. As detailed in “Section VHHs penetration and transport through barriers”, some VHHs have demonstrated the ability to cross the blood–brain barrier (BBB), offering enhanced potential for the diagnosis and treatment of brain cancer, particularly in cases where the BBB is disrupted [142,143,144,145,146,147,148]. A research study indicates that ^177^Lu-labeled anti-CD20 VHHs remain stable in human serum, with over 91% of the complexes intact 144 h post injection [149]. While they are stable, the low molecular weight of VHHs leads to rapid renal clearance. However, VHHs remain bound to the antigen for an extended period [150]. Supporting this, several studies have demonstrated that ^111^In-labeled anti-HER2 VHHs exhibit high specific uptake in HER2-positive brain tumors from 1 h to 3 d after injection. In contrast, ^111^In-labeled mAb Trastuzumab shows high nonspecific uptake in highly vascularized organs, such as the heart, spleen, and liver [147]. 

CDR3 regions play a crucial role in antigen binding by forming prominent grooves on their surface, while other sections of CDR3 can penetrate deeply into the active site of a lysozyme complex. These distinctive characteristics of the VHH fragment contribute to its higher affinity, solubility, and anti-aggregation properties [141,151,152,153,154].

### 4.3. Characteristics of VHHs to Develop Novel ADCs

Since their discovery, over 100 VHHs have been isolated, targeting areas relevant to oncology, in vivo imaging, hematology, and infectious diseases, as well as neurological and inflammatory disorders. VHHs are particularly well-suited for these applications because of their small size, target specificity, and long CDR3 loops, which help overcome many of the limitations associated with small-molecule synthetic drugs, such as limited specificity and off-target toxicity. These unique features of VHHs, compared to mAbs, present opportunities for developing VHH drug conjugates (nADCs or “nanoADCs”) with distinct pharmacological benefits (Table 2 and Appendix A). nADCs can potentially rival traditional ADCs due to their superior solid tumor penetration, enhanced stability, and their ability to significantly inhibit cancer cell growth [120,131,132,155,156]. A distinctive capability of VHHs is their ability to target epitopes in hard-to-reach locations that larger molecules, such as conventional mAbs, often cannot access. VHHs can bind in a ring-like fashion, allowing them to recognize epitopes typically inaccessible to standard antibodies, such as ion channel domains and intracellular proteins [113,120,133,134,135,136,137]. 

In summary, VHHs possess several ideal characteristics for drug conjugation, including high thermal and chemical stability, excellent solubility and strict monomeric behavior. Their small size (approximately 2.5 nm in diameter and 4 nm in length, M_W_: ~12–17 kDa) facilitates a better tissue penetration, while their relatively low production cost, ease of modification by genetic engineering means, format flexibility, low immunogenicity, and modularity further enhance their suitability for therapeutic applications [132,138,139].

#### VHHs’ Penetration and Transport through Barriers

In homeostasis, the BBB prevents conventional antibodies from crossing into the brain due to Fc-receptor-mediated efflux back into the bloodstream. The BBB’s permeability is restricted to receptor-specific ligands or molecules that are lipophilic and have a molecular weight under 400 Da, making it difficult to achieve therapeutic concentrations in the brain. Consequently, only 0.01–0.4% of blood proteins, including therapeutic antibodies such as IgG (150 kDa), can passively diffuse into the central nervous system (CNS) [145,157,158,159]. Some VHHs are described to naturally cross the BBB without external intervention by adsorptive-mediated transcytosis. This mechanism involves VHHs with a high isoelectric point (pI ~9.5) binding to anionic sites on endothelial cells, facilitating penetration. Additionally, cell-penetrating peptides can be fused to VHHs to enhance their ability to cross cell membranes and the BBB. Device-based and physicochemical methods, such as convection-enhanced delivery (CED), use microcatheters for direct brain infusion. Receptor-mediated transcytosis exploits specific BBB receptors, such as the transferrin receptor, for VHH transport via endocytosis and transcytosis [145]. Carrier systems like liposomes, extracellular vesicles, and nanoparticles can also be used to deliver VHHs across the BBB, increasing their bioavailability and effectiveness. Importantly, some pathological conditions like cancer and inflammation can compromise the BBB’s integrity, allowing mAbs and VHHs to enter the CNS [145,157,158,159].

The use of VHHs enables a more targeted payload delivery to less accessible areas for conventional antibodies [140,141,142,143,144,145,148,160,161], offering a solution to circumvent complications posed by barriers such as the BBB, blood–tumor barrier (BTB), and blood–synovial barrier. VHHs can be used in various ways, both alone and in combination. For example, administering antibodies against the receptor covering the first perivascular line of the tumor can saturate the blood–synovial barrier, and the VHHs would then be able to enter deeper areas where the stability of the ADCs is compromised, but not that of the nADCs [120,131,132,147,148,161].

### 4.4. The Plasticity of the VHH and the Opportunities for Conjugation

VHHs share many characteristics of mAbs that permit their conjugation to payloads, but their simpler structure facilitates genetic modification without losing their affinity. Conjugation can be performed using traditional protocols for lysine, cysteine or site-directed conjugation. Notably, the surface of VHHs is rich in amino acids such as lysine, aspartic acid, and glutamic acid, allowing for higher DARs (drug-to-VHH ratios) through conjugation at these residues [149,151].

The structure of VHHs and their lack of complex post-translational modifications allows for the introduction of non-natural amino acids or cysteines for site-specific conjugation when compared to IgG antibodies. Site-directed conjugation is preferred, as the presence of lysines in the CDR binding sites may compromise their affinity if a drug is conjugated via these sites. Introducing an additional cysteine at a location distant from the paratope, preferably at the C-terminal, can partly solve these issues. Other effective conjugation alternatives include sortase A, transglutaminase, and GTPase enzymes, although they require scaffold modification to be site-specific [141,151,152,153,154].

In clinical settings, unconjugated antibodies are often well tolerated, allowing for high doses that saturate receptors on the cell layers closer to the blood vessel and enable deeper tumor penetration. However, the payload toxicity of ADCs limits the dosage and frequency of administration, which can restrict tumor penetration, allowing for regrowth between doses (typically administered every three weeks in current therapies). Thus, designing treatment strategies that enhance tumor penetration could lead to greater efficacy and improve clinical success rates for ADCs and other protein–drug conjugates [7,11,162,163]. In addition to binding affinity, antibody size influences tumor penetration. Decreasing the size of a conjugate while maintaining affinity and specificity facilitates its entry into solid tumors through blood vessels, significantly enhancing its therapeutic effect. As an example, comparative studies in patient-derived organoid (PDO) models between an anti-5T4 VHH-SN38 and a conventional anti-5T4-SN38 have demonstrated superior penetration and greater tumor regression by the VHH [164].

Moreover, unlike conventional antibodies which may have prolonged circulation due to interaction with FcRn-mediated receptors, VHHs do not interact with FcRn, ensuring that their transport to lysosomes is not compromised [165,166,167]. This characteristic can lead to more effective payload release following lysosomal degradation. Finally, chemical modifications to ADCs can impair stability, making them more sensitive to changes in the intratumoral environment, leading to instability and the potential loss of interaction with the paratope. VHHs, however, are stable under adverse conditions that help maintain their functional integrity and effectiveness in tumor cell elimination [168,169,170,171,172,173,174,175,176,177,178,179,180,181,182,183,184,185]. 

### 4.5. VHHs as Carriers in Antibody–Drug Conjugates (nADCs)

To date, no single class of antibodies perfectly combines several ideal properties for targeting the moieties of ADCs, including low immunogenicity, rapid distribution, the fast clearance of unbound molecules, and high tumor accumulation. Despite substantial interest in ADC therapies, high failure rates have been observed in the clinical setting [186]. Effective tumor accumulation is critical for the antitumor activity of ADCs, especially against solid tumors, and depends on both tumor penetration/retention and antibody pharmacokinetics (Figure 5). Consequently, smaller recombinant antibody fragments such as VHHs and fragment of antigen binding (Fab) of mAbs have been explored as alternatives to intact mAbs for generating ADCs, but their clinical applications have been limited by poor structural stability. Recently, a class of small, soluble antibodies derived from HCAbs such as VHHs have gained significant attention, due to their minimal molecular size and high thermal stability [186,187,188,189,190,191] (Appendix A). Along with their high specificity, their solubility, low immunogenicity, and ease of production make VHHs ideal for constructing “nano-ADC” (nADC).

The small size of VHHs facilitates excellent and rapid tumor penetration in vivo. Additionally, this small size is crucial for our design, as it enables VHHs to bring the nADCs construct very close to the membrane after binding to the surface target [113]. This capability makes nADCs a promising approach for next−generation targeted drug conjugates [172,192,193,194,195,196,197,198,199]. Furthermore, their smaller size and aqueous solubility enable quicker tumor infiltration compared to mAbs. VHHs’ superior specificity arises from their ability to bind to epitopes that conventional mAbs cannot reach. Studies indicate that VHHs achieve higher tumor-to-background ratios in molecular imaging in vivo, due to their precise binding and accumulation in tumors, along with the rapid clearance of unbound constructs, resulting in lower background signals and reduced toxicity. Importantly, VHHs exhibit exceptional versatility, making them ideal for integrating various functional modules and enhancing the clinical potential of nADCs. Overall, VHH−drug conjugates or nADCs are emerging as a promising alternative to conventional ADCs [143,164,192,200,201,202,203,204,205,206,207,208,209,210,211,212].

Building on these advantages, VHHs also demonstrate good performance in imaging applications. Their ability to achieve favorable target-to-noise ratios is evident in research settings where VHHs conjugated with radionuclides are used for cancer imaging. For instance, single-positron emission tomography (SPECT) or positron emission tomography (PET) combined with micro-computed tomography (micro-CT) has effectively utilized VHHs for imaging primary tumors and metastatic sites across various cancer models, including melanoma, breast cancer, ovarian cancer, early pancreatic lesions, and advanced pancreatic ductal adenocarcinoma. PET/CT imaging with VHHs provides excellent clarity and signal-to-noise ratios [150,156,173,213,214,215,216].

The advent of new technologies that increase the bioavailability of VHHs will facilitate their transition into clinical use. The main methods include:Reduced glomerular filtration rate due to increased glomerular mass or hydrodynamic radius. Strategies such as combining VHHs with nanoparticles and liposomes, or modifying them with polyethylene glycol (PEG), are proving effective in enhancing drug delivery to cancer cells. These approaches improve penetration into solid tumors and reduce systemic toxicity [217,218,219,220,221].Binding to plasma proteins with extended half-life. To extend the half-life of VHHs and improve their efficacy, methods are being developed to optimize their affinity and release dynamics. Advances include fusing VHHs with human serum albumin (HSA) or the Fc domain. Other strategies include the “fenobody” platform developed by Kelong Fan and colleagues, where VHHs targeting the H5N1 virus were displayed on a 24-subunit ferritin oligomer. By replacing ferritin’s fifth helix with the VHH, affinity and half-life were significantly improved, offering substantial advantages for large-scale biotechnological applications and promoting the broader adoption of VHH technology [220,222,223,224,225,226].Structural and design modifications. Enhancing VHH efficacy against tumor antigens involves techniques such as forming CDR rings, stabilizing secondary structures, or creating bispecific and multispecific VHHs to improve affinity, specificity, stability, and solubility in challenging physiological environments [227,228,229,230,231,232].

### 4.6. Advances in the Development of nADCs

To date, no VHH–drug conjugates have entered clinical research, and all published studies remain at the preclinical stage. These studies have explored the direct conjugation of the nanoantibody to drugs using multiple forms and linkers, and in some cases, VHHs have been used in conjunction with other formulations to deliver the drug. The main conjugated payloads include doxorubicin, MMAE, cisplatin, and SN38 (Table 3). While nADCs show promise in terms of therapeutic response, a thorough analysis suggests that traditional cytotoxic agents may not be the most suitable choice for advancing their clinical development. Matching the right payload with the appropriate linker and antibody formats is crucial for optimizing nADC efficacy. More potent payloads, such as topoisomerase I inhibitors, could enhance tumor penetrability. Although antimicrotubule inhibitors have shown promising activity in preclinical studies with lymphomas and leukemias, their effectiveness is generally lower in solid tumors compared to antitopoisomerase inhibitors. New payloads, such as pyrrolobenzodiazepines, may be the most suitable for nADCs due to their high potency. Regarding the limitations caused by toxicity [233], combining PBDs with VHHs might reduce toxicity, given VHHs’ shorter plasma retention time. Furthermore, the enhanced penetrability of VHHs in solid tumors compared to conventional antibodies could help maintain a strong antitumor response.

### 4.7. Disadvantages of VHH for nADCs and Possible Improvements

#### 4.7.1. Fast Clearance and Renal Retention

One of the main disadvantages of using single-domain antibodies or other small protein scaffolds, compared with traditional mAbs, resides in their rapid renal clearance [219]. The short half-life of VHHs in blood circulation is advantageous for imaging applications, where rapid clearance allows for a faster visualization and optimization of images while minimizing toxicity. However, this rapid clearance can compromise therapeutic actions that require a more prolonged presence in the bloodstream. 

Of note, it has been shown that the number of polar residues in the C-terminal amino acid tag (e.g., poly-Histidine tags) significantly contributes to the kidney retention of VHHs. This is particularly relevant in preclinical trials, where different tags are used for antibody purification. Furthermore, these tags can be immunogenic. Although many clinically promising manufactured proteins are His-tagged, there is public concern about using this type of tag. Both the FDA and the European Medicines Agency (EMA) discourage its use due to potential undesired immune responses, despite the lack of official public specifications [247].

Kidney accumulation is not desirable for therapies that may inadvertently concentrate the carried toxicant in the glomerulus, leading to consistent local action. Therefore, multiple strategies have been investigated to reduce renal retention. Despite their quick diffusion through the vasculature, ability to deeply penetrate tumors, and excellent affinity, these factors can potentially offset the need for an extended half-life. In preclinical studies, VHHs have demonstrated superior tumor responses compared to conventional ADCs targeting the same antigens, even without the need for repeated dosing [217,219,248].

The short half-life can be addressed with additional formulations, as described above, including PEGylation, fusion to serum proteins, multimerization, or fusion to an IgG Fc-domain, etc. Different administration strategies, such as continuous infusion devices or repeated doses, can also be employed [248,249,250,251,252,253].

A novel VHH that binds to serum albumin of different species has been discovered in native VHH libraries, facilitating its assembly into bispecific and multispecific antibodies to prolong the pharmacokinetic profile of the molecules [254]. For example, a bivalent anti-VEGF VHH demonstrated a 1.8-fold longer half-life compared to the monovalent form. PEGylation resulted in a 12-fold increase in the half-life of an anti-CEA/CD3 bispecific VHH [238,253,255,256,257,258,259,260,261]. Clinical proof-of-concept of the extended half-life achieved in this manner has been demonstrated for an anti-IL-6R and anti-TNF VHH, fused to a serum albumin-binding VHH used in treating rheumatoid arthritis [262].

The kidney retention of radiolabeled VHHs can be significantly reduced by using an untagged C-terminus in conjunction with the plasma expander gelofusine [263]. VHHs produced without the C-terminal His-tag (a common production method) exhibit a 60% reduction in kidney uptake compared to tagged VHHs [264,265]. Specifically, untagged VHHs show a 70% decrease in kidney accumulation compared to Myc-His-tagged VHHs, and co-infusion with gelofusine results in a 90% reduction in kidney accumulation [263]. The administration of isotypes or “cold” VHHs, as they are also known, can be an attractive dual strategy as they can be used to cause initial unspecific renal retention, leading to improved clearance of nADCs. Recently, conjugation strategies using enzymes such as sortase or transglutaminase have been shown to concurrently remove or obliterate the need for purification tags, such as the His-tag. This results in VHHs that are less likely to be retained in the kidneys [149,266,267,268].

Rapid clearance requires repeated drug applications, which may affect the development of therapeutic responses and patients’ quality of life. The ADCs approved to date are administered at intervals of several weeks, but this schedule is also influenced by the pharmacokinetics of the antibodies used and the toxicity associated with their use. This schedule is also influenced by the pharmacokinetics of the antibodies used and the associated toxicity.

#### 4.7.2. Is the Rapid Clearance of VHHs Necessarily a Disadvantage?

ADCs are revolutionizing cancer treatment. Their mechanisms of action and associated toxicities, which are unprecedented compared to other therapies, challenge traditional dogmas about bioavailability. The variability in responses to the same ADC depends on the tumor type. The occurrence of severe toxicities such as neutropenia, keratitis, interstitial lung disease, and even fatalities, highlights the need for a better understanding of these drugs [269,270,271,272]. Many toxicities have been linked to the instability of the compound in the bloodstream, its off-target effects, and nonspecific interactions with blood cells, as previously mentioned. We propose that rather than it being a problem, the rapid vascular permeability and distribution to tissues, swift and deep tumor penetration, and clearance of VHH-based ADCs or nADCs could offer an opportunity and could be leveraged as a strategy to reduce toxicity.

### 4.8. Final Consideration: Improving the Efficacy in Solid Tumors ADCs versus nADCs: Strategies to Overcome Major Barriers

The uptake and penetration of mAbs in tumors are constrained by several tumor-specific pathophysiological factors, including elevated interstitial fluid pressure, a dense extracellular matrix, and an aberrant vascular network. These limitations stem from the inherent design of an ADC, which targets an overexpressed and rapidly internalizing antigen on tumor cells. The impact of these barriers may be more pronounced for ADCs compared to conventional mAbs, due to the lower doses typically used in clinical settings [273,274,275].

The delivery of biologicals to the CNS presents a significant challenge, primarily due to the presence of various barriers separating the CNS from the periphery. These barriers include the BBB, acting as a highly selective and regulated filter that tightly controls the passage of substances between the bloodstream and the brain parenchyma (Figure 6). Only ~0.1% of circulating macromolecules can cross the BBB, severely limiting the use of biologics in treating CNS-related diseases. In a state of homeostasis, the integrity of the BBB makes it difficult for conventional antibodies to cross spontaneously, largely due to Fc-receptor-mediated efflux back into the bloodstream. Consequently, the transfer of biopharmaceuticals across the BBB remains a significant obstacle in developing therapeutics targeting the CNS. This challenge is exacerbated in the case of ADCs, where the antibodies not only carry a payload but also undergo modifications that affect their structure, size, and interaction with the endothelium. Instability within the CNS, resulting in premature drug release before reaching the target tumor, can lead to increased neurological toxicity and severe complications [145,157,276].

The blood–tumor barrier (BTB) refers to the physical and biological barriers between the circulatory system and solid tumors. Analogous to the BBB, the BTB is observed in tumors located outside the central nervous system. Endothelial cells lining the blood vessels within the tumor often display abnormal organization and altered intercellular junctions. Additionally, the tumor microenvironment features a dense and disorganized extracellular matrix, which poses challenges to the penetration of therapeutic agents [277]. While blood vessels in tumors may exhibit increased permeability compared to normal vasculature, this permeability is heterogeneous and does not necessarily ensure uniform drug distribution. Furthermore, tumors typically exhibit high interstitial pressure, which may hinder the vascular permeability of therapeutic agents from blood vessels into the tumor mass [278,279].

A very high affinity of an antibody can result in restricted tumor penetration and heterogeneous tumor distribution, with preferential binding of the antibody to tumor cells localized around tumor vasculature. The binding site barrier (BSB) hypothesis, originally proposed by John N. Weinstein [280], explains the non-uniform distribution of mAbs in tumor nodules [281,282,283]. He suggested that cell populations near the blood vessels with high antigen density and binding affinity likely elicit a strong BSB. Multiple solid tumor systems, such as pancreatic ductal adenocarcinoma (PDAC), non-small cell lung cancer (NSCLC), aggressive urothelial carcinoma, and some breast cancers exhibit this pattern, revealing tumor-associated fibroblasts as a major component of the BSB [284,285,286].

These barriers cause ADCs to remain in the surrounding tumor vasculature, preventing their distribution to central tumoral areas. Despite the potential of mAbs to better permeate cancer than normal tissues due to the typical leaky tumor vasculature, their large size poses a challenge to efficient solid tumor treatment, explaining the prevalent early application and clinical success of ADCs in hematological malignancies. VHHs, with their ability to penetrate deeply into tumors much faster than conventional antibodies and their stability at low pH, make them ideal for crossing the described barriers and reaching tumor areas that are normally niches for cancer stem cells and largely resistant to conventional therapies [6,41,84,186,283,284,287,288,289].

Although no such small-format conjugates have yet reached market approval, encouraging preclinical and preliminary clinical results hold promise for nADCs for the future implementation of these smaller formats in the clinical arsenal.

## 5. Conclusions

Single-domain antibodies offer unique advantages, including small size, high stability, specificity, ease of production, and low immunogenicity, positioning them as a promising tool for future ADC constructs. Dose-limiting hematologic toxicities, particularly thrombocytopenia and neutropenia, represent some of the most serious adverse events commonly associated with approved ADCs. These toxicities are primarily attributed to antigen-independent off-tumor targeting, which can arise from several mechanisms. The uptake of intact ADCs into normal cells can occur through nonspecific endocytosis or via internalization after binding to the target antigen or Fc/C-type lectin receptors. In the extracellular fluid, payloads released from ADC deconjugation or from apoptotic cells—both targeted and non-targeted—can also enter normal cells. This entry can happen through passive diffusion for membrane-permeable payloads or nonspecific endocytosis for membrane-impermeable linker–payload complexes [23,290]. VHHs offer a promising solution to mitigate these issues due to their small size, high stability, specificity, ease of production, and low immunogenicity. Their unique properties make VHHs a valuable tool for improving the safety and efficacy of future ADC constructs.

While VHHs remain bound to the antigen for an extended period, a significant challenge with VHH-based ADCs (nADCs) could reside in their rapid renal clearance, which could require high and frequent dosing. While various techniques have been explored to increase the VHH half-life extension in the bloodstream, their effectiveness in enhancing the efficacy or safety of nADCs through prolonged exposure remains inconclusive and warrants further research. Addressing these challenges could potentially improve the pharmacological and safety profiles of nADCs, enabling the more precise and effective targeting of cancer cells.

## 6. Limitations of the Study

The study’s limitations include the restricted timeframe from 1 January 2014 to 30 June 2024, which may exclude other relevant studies published before or after this period. There is a language bias, as only articles in English and Spanish that are available online were included, thus missing studies published in other languages. The study relies on specific databases (PubMed, Cochrane Library, ScienceDirect and LILACS), potentially overlooking relevant studies from other sources. The broad inclusion criteria may lead to varying levels of evidence and quality among the selected studies. There is also a potential for publication bias due to the inclusion of only online articles. The findings may not be generalizable to all ADCs. These limitations should be considered when interpreting the study’s conclusions.

## Figures and Tables

**Figure 1 cancers-16-02681-f001:**
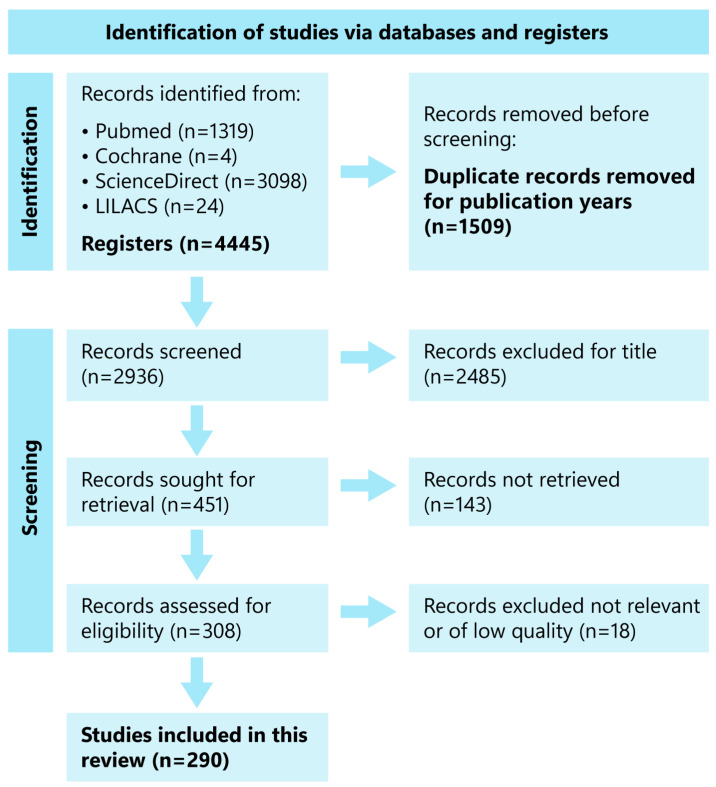
Strategy for identification, screening and selection of articles for systematic review.

**Figure 3 cancers-16-02681-f003:**
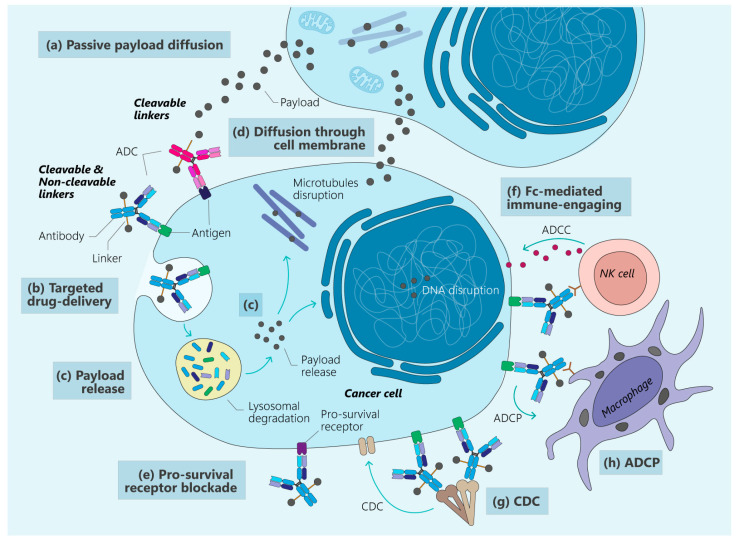
ADC mechanism of action. (a) Passive payload diffusion. The drug that has been released from the antibody before binding to the antigen passively diffuses into the surrounding cells. (b) Targeted drug-delivery. The ADC binds to the antigen in the cell and is internalized by endocytosis. Once the conjugated antibody binds to target cells, it can be internalized via endocytosis, carrying along the payload. (c) It is degraded in lysosomes, releasing the drug that carries out its intracellular action. Once bound to the target cells or inside the cell, the conjugate can release its payload (e.g., a cytotoxic drug). This may lead to cell death. (d) Diffusion through the plasma membrane with action on surrounding cells. (e) Antagonism or agonism of the target receptors. The antibody is designed to specifically recognize and bind to the antigens present on the surface of certain types of cells, such as tumor cells. Antibody–drug conjugates can have agonistic or antagonistic effects when binding to their specific receptor on the cell surface, allowing them to modify intracellular pathways and potentially inhibit cell growth or metabolism, depending on their design and specific cellular context. (f) Some antibody–drug conjugates are engineered not only to directly target specific cells by binding to their surface receptors but also to harness the immune system for enhanced cell destruction. This can occur through mechanisms such as (g) antibody-dependent cell-mediated cytotoxicity (ADCC), where immune cells like Natural Killer (NK) are activated to recognize and kill the targeted cells. Additionally, antibody-dependent cellular phagocytosis (ADCP) involves immune cells engulfing and digesting the marked cells, further contributing to their elimination. (h) Complement-dependent cytotoxicity (CDC) is another mechanism employed, where the conjugates activate the complement system to induce cell lysis. These strategies collectively bolster the therapeutic efficacy of antibody–drug conjugates by leveraging immune responses to eliminate target cells more effectively.

**Figure 4 cancers-16-02681-f004:**
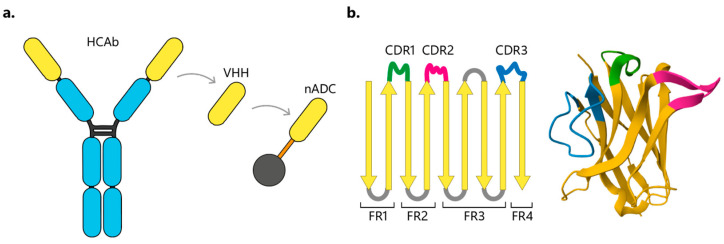
Structure of VHH and nADC. (**a**) Schematic representation of an HCAb, a VHH and a nADC. (**b**) Schematic representation of the VHH structure and molecular 3D structure (PDB reference: 1I3V), where the complementarity determining regions (CDRs), known to be responsible for antigen recognition, are displayed (CDR1, in green; CDR2, in pink; CDR3, in blue).

**Figure 5 cancers-16-02681-f005:**
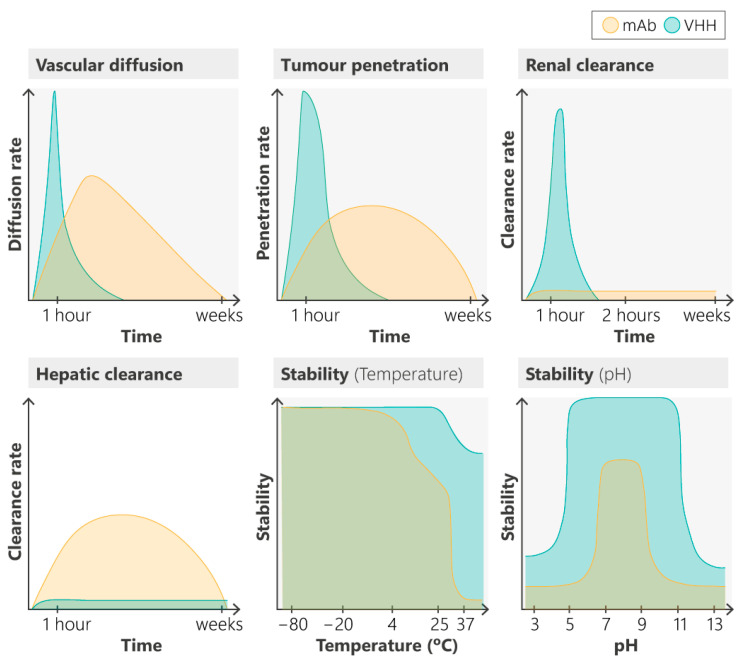
Graphical comparison of the properties of a conventional mAb versus VHH, including the vascular diffusion, tumor penetration, renal and hepatic clearance rates, and the stability under storage at a wide range of temperature conditions and stability at different pH.

**Figure 6 cancers-16-02681-f006:**
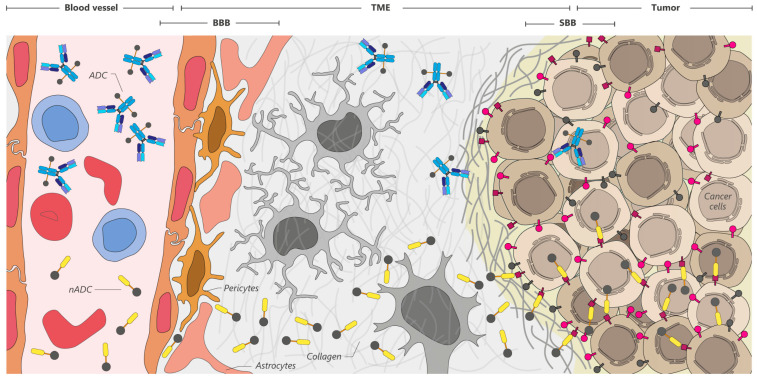
The transport of ADCs and nADCs across the blood–brain barrier (BBB) and the binding site barrier (BSB). The BBB restricts the movement of substances from the bloodstream into the brain parenchyma, whereas the BSB limits the penetration of antibodies (mAbs) and VHHs into tumors, leading to the uneven distribution of ADCs and nADCs. Compared to mAb-based ADCs, nADCs exhibit enhanced ability to traverse both the BBB and BSB, facilitating more effective delivery to brain and tumor tissues. TEM: Tumor microenvironment.

**Table 1 cancers-16-02681-t001:** Main characteristics of 1st, 2nd and 3rd ADC generations.

ADC Generation	Antibody	Payload Potency	Plasma Stability	Homogenous DAR	Toxicity	Off-Target Action
1st	Murine	+	+	+	++++	+++++
2nd	Humanized	+++	+++	+++	++	++
3rd	Fully human	+++++	++++	++++	+	+
Future generations	Improvement in all components					

**Table 2 cancers-16-02681-t002:** Major differences between mAbs and VHH.

	mAb	VHH
Size	~14.5 nm	2.5 to 4 nm
MW	150 kDa	12–17 kDa
Antibody production	Mammalian cell post-translational modification needed	Mammalian or microbial, naked and no post-translational modification needed
Immunogenicity and complexity	High glycosylation and interactions with immune cells via Fc/FcR	Low, no Fc/FcR interaction
Stability	More dependent on pH and temperature. Aggregation with other proteins	Wide pH range, extreme chemical and thermal stability. Low aggregation
Clearance	Hepatic, long half-life	Renal, relatively short half-live
Tissue penetration	Low	High tissue permeability, can cross the BBB
Epitope recognition	Difficult recognition of hidden sites	Strong, with a site that cannot be reached by normal antibodies
Production cost and standardization	High	Relative low
Humanization and structural modification	Can lose function or stability	Easy modification
Affinity	nM-µM	pM-nM

**Table 3 cancers-16-02681-t003:** Preclinical studies of VHH–drug conjugates, where CEACAM5: Carcinoembryonic Antigen-Related Cell Adhesion Molecule 5, PSMA: Prostate-Specific Membrane Antigen, HER2: Human Epidermal Growth Factor Receptor 2, EGFR: Human Epidermal Growth Factor Receptor 1, VEGFR2: Vascular Endothelial Growth Factor Receptor 2, NHS: N-hydroxysuccinimide ester, EDC: 1-ethyl-3-(3-dimethylaminopropyl)carbodiimide hydrochloride), aMHC-II—Major Histocompatibility Complex Class II.

VHH	Target	Payload	Cancer/Cell Line Models	Linker	Method of Conjugation	Ref.
Anti-CD22-VHHs	CD22	DM1	Lymphoma	Succinimidyl trans-4-maleimidylmethyl cyclohexane-1- carboxylate (SMCC)	Maleimide	[234]
n501-SN38	Oncofetal antigen 5T4	SN38	Solid tumor (Pancreas, Breast, Ovarian, Colon)	ClA2	Maleimide	[164]
B9-S84C	CEACAM5	Maytansinoid DM4	Solid tumor (Pancreas)	MC-VC-PAB	Maleimide	[235]
Nb 11-1	CD147	Doxorrubicine	CD147-positive tumors	-	Maleimide	[236]
VH1-HLE, VH2-VH1, VH2-VH1-HLE, and J591	PSMA	DNA-alkylating agent (DGN549) indolinobenzodiazepine DNA-alkylating monoimine	Prostate cancer CWR22Rv1 DU145 and DU145-PSMA cell lines	-	Maleimide	[194]
NB7	PSMA	Doxorrubicine	Prostate cancer PC3-PIP and PC3-flu	pH-sensitive linker N-(β-maleimidopropionic acid) hydrazide (BMPH),	Maleimide	[237]
VHH7	aMHC-II	DM1	Lymphoma	-	Sortase-mediated site-specific protein engineering	[173]
HuNbTROP2-HSA	TROP2	MMAE	Pancreatic cancer	MC-VC-PAB,	Maleimide	[238]
VH-Fc 3C9	Mesothelin	MMAE	Solid tumor	VC-PAB	Maleimide	[239]
Tetravalent biparatopic anti-EGFR VHH–drug	EGFR	MMAE	Solid tumor	MC-VC-PAB	Maleimide	[240]
2Rs15d	HER2	Duocarmycin	HER2 positive tumor	Compound S22 Synthetic duocarmycin linked to Psyche	VHH fused to Cupid protein Psyche-duocarmycin	[241]
PEGylated-antiEGFR VHH	EGFR	Pt(IV) (prodrug of oxaliplatin)	EGFR positive cell lines	Mal-Pt(IV)	Transglutaminase (mTGase) mediated ligation	[242]
11A4	HER2	Auristatin F (AF)		platinum-based Lx linker	Maleimide	[217]
VHH-conjugated H40-PEG	VEGFR2	Methotrexate	HEK293 (human embryonic kidney cells) Breast cancer KDR293 (overexpressed for VEGFR2 receptors)	NHS/EDC	Random lysines	[243]
scPDL1-DM1	PDL1	DM1	PDL1 positive cells	Succinimidyl trans-4-maleimidylmethyl cyclohexane-1- carboxylate (SMCC)	Maleimide	[244]
N, 7D12-9G8	EGFR	Cisplatin	A375, A431, Solid tumors	Mal-pt	Maleimide	[245]
Single-chain anti-HER2	HER2	Doxorubicine	BT474-M3, NCI-N87	N-[α-(2-[N-maleimido]propyonylamido)-PEG-omega-oxycarbonyl]-DSPE	Maleimide	[246]

## Data Availability

No new data were created or analyzed in this study. Data sharing is not applicable to this article.

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
