# Peer review of "Single-Domain Antibodies as Antibody–Drug Conjugates: From Promise to Practice—A Systematic Review"

_cancers, 2024, doi:10.3390/cancers16152681_

Round 1

Reviewer 1 Report

Comments and Suggestions for Authors

The manuscript titled "Single domain antibodies as antibody-drug conjugates: from promise to practice. A systematic review" provides a comprehensive overview of the potential of single-domain antibodies (VHHs or nanobodies) as a new class of antibody-drug conjugates (ADCs). The review is well-structured, covering various aspects including the history, properties, advantages, and current state of VHH-based ADCs. The authors have effectively highlighted the challenges and opportunities in this emerging field. However, there are several areas that need further clarification, elaboration, and improvement.

1. Abstract:

   - The abstract is concise but could benefit from more specific details about the key findings and conclusions. Consider including more quantitative data to highlight the significance of VHHs in ADCs.

2. Introduction:

   - The introduction provides a good historical context but lacks a clear statement of the review's objectives. It would be helpful to explicitly state what this review aims to achieve and what gaps in the current literature it addresses.

3. Systematic Review Methodology:

   - The methodology section could be expanded. While the PRISMA guidelines are mentioned, details about the search strategy, inclusion/exclusion criteria, and the process of data extraction and synthesis are sparse. Providing a flowchart of the review process would enhance transparency.

4. VHHs as ADCs:

   - The section on VHHs as ADCs is informative but could be more detailed. Specifically, the advantages of VHHs over traditional antibodies should be supported by more quantitative data and comparative studies. Tables summarizing the properties, advantages, and challenges of VHHs versus traditional antibodies would be useful.

5. Figures and Tables:

   - Figures 2 and 3 are mentioned but not adequately described in the text. Ensure that all figures and tables are referenced and discussed in detail. Additionally, some of the figures are quite complex and could benefit from more detailed captions explaining each part of the figure.

6. Challenges and Future Perspectives:

   - The discussion on challenges faced by VHH-based ADCs is comprehensive. However, the future perspectives section could be more forward-looking. Consider discussing emerging technologies or novel approaches that could address current limitations.:

7. Site-Specific Conjugation:

   - The manuscript would benefit from discussing recent advancements in site-specific conjugation techniques, particularly the use of affinity molecules such as AJICAP and AJICAP-M, which are becoming more prominent. Adding a section that elaborates on these techniques and their advantages in ADC development would provide a more comprehensive view. Appropriate citations should be included to support this discussion.

8. Payloads:

   - While the manuscript covers cytotoxic payloads extensively, it should also address ADCs with non-cytotoxic mechanisms of action, often referred to as "Novel format conjugates." These ADCs represent an important advancement in the field and should be discussed in detail, along with relevant examples and citations to recent studies.

9. Table 2:

   - Table 2 compares antibodies and VHHs, but it would be beneficial to include VHH-Fc (HC-Ab) in the comparison as well. This addition would provide a more complete overview of the different formats of antibodies and their respective advantages and disadvantages.

Reviewer 2 Report

Comments and Suggestions for Authors

This review presents the VHH domains as potential target-recognition elements of ADCs. Regarding the particular antigen recognition mode, small size and pharmacokinetic properties, in which VHH differ from full-length antibodies, but also their high stability, this is a valuable topic that could expand the therapeutic options involving drug conjugates in the future. A large body of literature has been covered.

My main comment on the review is that it is too eclectically written and not well organized, as the sentences and statements repeat (please see the remarks below), and the topics often miss a logical thread. It is also very long, which I am not sure is an intention of the authors, and especially the introduction recapitulating the general methods for design of ADCs goes into great length. To that, several valuable strategies, which indeed represent milestones in ADC development, are briefly mentioned and pooled into general statements, with little systematics used. This background knowledge could be summarized as one paragraph (=state of the art), followed by a summary of VHH properties beneficial when they are used as stand-alone proteins for therapeutic purposes, and then the core of the review should be the overview of the attempts to derivatize them into nADCs, and challenges or advantages posed when they are used in this way. Targeting BBB is also important and deserves special attention, so it could be one subchapter here. Optimally, the article should be concluded with a summary of future prospects in nADC development and use.

Figures should be corrected, paying more attention to detail regarding ADC mechanism and potential advantages of VHHs as ADCs.

Finally, the language used within the review is at present not OK, scientific expressions are often not correctly rendered, and I strongly recommend a careful re-read and correction by all authors, because different text passages here are clearly at different levels of sentence organization.

Please find below a list of remarks which I hope you will find helpful.

Lines 39-40: the mentioning of caplacizumab is better suitable for the introduction part as for the abstract

Figure 1: identification (typo)

Line 131: two independent investigators (VMMP, MB and AJS) – this is 3 investigators

Line 162: should be rapidly released upon entry into lysosomes

Line 164: “higher toxicity than other drugs” – please be more specific (usually, chemotherapeutics are compared here)

Line 181: “the mode“ – too general. Is this the mode of cleavage? Or the mode of coupling (chemical, site-directed, enzymatic)?

Line 184: “site directed linkage to specific amino acids and enzymatic approaches”- coupling can be chemical or enzymatic, and the same holds true for chemical or enzyme-mediated approaches

Line 190: “other chemical approaches to incorporate non-natural amino acids by genetic engineering into mAb”: non-natural amino acids are mostly incorporated biologically (in vivo) and then derivatized chemically.

Line 194: “enzymatic strategies „ - “enzyme-mediated strategies”. Of these, glycoengineering is a combined strategy because the glycans are enzymatically cleaved, and then chemically derivatized.

Lines 215-217 are repeated from the paragraph above.

Line 220: “low drug capacity per antibody” – what is a low drug capacity per antibody? Is this paragraph on chemotherapeutics or on antibodies?

Line 255: “receptor-mediated endocytosis“ – logical connection to the previous paragraph is missing

Line 257: “Once in the cytoplasm“ – this sentence refers to the released toxin

Line 267: this paragraph has reversed order of the sentences: antibody related functions directly involve innate or complement immune effectors, such as Fc-mediated killing mechanisms, etc.

Figure 3 requires improvement. Cleavable linkers are shown to lyse on the surface of the cancer cells, while they are all designed to be preferentially cleaved inside the cell; NK cells and macrophage appear to react with an entity fused to the end of the antibody Fc, and complement components are drawn to react with antibody CH3 domains. Neighbouring bystander cells are not visible.

Line 307: “instability of the binding” – of the drug to the antibody?

Line 323: “antibody-tumor ratios“ – antibody to drug ratios

Table 1: “DAR homogenization“: should be homogenous DAR, which relates to a well defined antibody preparation

Line 350: “extravasating at the tumour site” - Extravasation is a rare complication from intravenous chemotherapy administration and very rare for antibodies, and a specialized term not to be used in this context

Line 352: “serum half-life of 2-3 weeks” – tis is only for IgG1, and typically shorter for ADCs. I propose omitting the entire next paragraph, because for the ADC efficacy the linker is supposed to be degraded and the drug released, regardless of the antibody recycling.

Line 370: “without losing significant Ag-binding capacity” – without significantly losing the Ag-binding capacity.

Line 373: "induce relatively low immunogenicity” – induce mild immunogenic response, immunogenicity is the property of the drug

Line 379: ending in: “intratumoral injection” – references missing

Line 384: than in the conventional VH domains in human IgG

Lines 392-396: these are examples of few studies VHH scaffolds. It would be better to describe the thermostability of VHH in more general terms, and not refer to individual case studies.

Lines 445-448: This paragraph describes structural features of the VHHs and does not fit into the context here. I guess that what the authors wish to say is that the extended CDR3 loops of VHHs make the interactions with grooves on antigen surface possible, or penetrate within the active sites of enzymes.

Line 452: …”with some in advanced stages of clinical trials” , while some have already reached advanced stages of clinical trials and up to date, already multiple molecules have been approved for human use

Line 460: “Agic epitopes“ – antigenic?

Lines 460-464 describe (better) what is written in the lines 445-448.

Table 2: many typos: posttranslational, glycosylation, aggregation, half-life, cryptic sites can be reached (if this is what “site tan con not be reached “ means. Other issues: also most therapeutic antibodies have Tms > 60 °C, and the affinities of antibodies are surely comparable with VHHs

Line 497: “sortase A, transglutaminase, and GTPase enzymes” – these are conjugation methods which also require scaffold modification to be site-specific, please complete the sentence.

Line 517: “very low pH or high urea concentrations” – the statement that VHHs are stable under these conditions makes sense, only the previous paragraph talks about changes in the intratumor environment, please make a logical connection.

Line 527: Figure 5: Stability (temperature) clearly refers to Storage stability, and the thermal atbility discussed in the text is not mentioned

Line 564: show promise

Line 468: TopoI

Table 3: “Linker or method of conjugation” – is not OK. Please separate by conjugation method (or target: cysteine, lysine…) and by linker. Many typos: lysine, breast cancer, HEK293 cells are surely not a tumor model, doxorubicin, lymphoma, maleimide, Val-Cit is sometimes vc, sometimes VC, etc. reference 197: anti-Her2 VHH is fused to Cupid, which forms a complex with Psyche and Psyche is labelled with duocarmycin. Reference 199: urease is an enzyme and this is not a classical payload for ADCs.

Line 603: “…, including PEGylation, fusion to serum albumin, multimerization, or fusion to an IgG Fc-domain” – please order by the strategy of halflife extension and briefly explain it.

Lines 609: reference missing

Line 625: “promise in removing production tags „ – concurrently removing or obliterating the need for purification tags

Line 628: “The approved ADcs to date „ – the ADCs approved to date

Line 641: offer an opportunity

Line 677-679: the comments on high affinity are very relevant, only the high affinity of VHHs then should not be explicitly mentioned as an advantage

Line 701: Figure 6: This is the BBB barrier and not just any “biological barrier”.  What is SBB, BSB (binding site barrier) discussed above? The abbreviations should be explained within the Figure Legends. The concept of the Figure is good, but it is not informative in elucidating the difference between mAbs and VHHs as ADCs.

Reviewer 3 Report

Comments and Suggestions for Authors

This paper "Single domain antibodies as antibody-drug conjugates: from 2 promise to practice. A systematic review", is interesting and can be accepted after major suggestion. 

1. I would appreciate if the title can be restructure using the words already given in the title. The current title though good, but not in a good flow. 

2. Please add specific lines in the abstract section about the methods and procedures applied. 

3. In the introduction section, the study rationale is not properly provided and supported by recent literature. 

4.  section 2.2 must discuss the research article type. Also, why are case reports, editor letters, and conference studies not included?

5. Why was the specific period (January 2014– June 24, 2024) preferred? why not those before?

Comments on the Quality of English Language

Minor editing of English language required

Round 2

Reviewer 1 Report

Comments and Suggestions for Authors

The authors have addressed almost all comments. Their efforts are much appreciated. However, the reviewer found a typographical error: reference 38 is incorrect. Please cite the appropriate literature (Org. Lett. 2024, 26, 27, 5597–5601).

Author Response

Comment 1: The authors have addressed almost all comments. Their efforts are much appreciated. However, the reviewer found a typographical error: reference 38 is incorrect. Please cite the appropriate literature (Org. Lett. 2024, 26, 27, 5597–5601).

Response 1: We thank Reviewer 1 for his comments and suggestions. Reference 38 is now updated.

Reviewer 2 Report

Comments and Suggestions for Authors

The authors have corrected the manuscript, which is now better organized, more systematically written, and the use of scientific terms is improved. They have also improved the quality of the Figures, which now better support the text. The only major remark I would have is that they mention Table S1, but it is not contained within the main text or in the Supplementary Files.

Please find below a short list of remarks:

Line 339: Prolactin receptor (PRLR): the abbreviation needs to be explained

Lines 379-380: Figure panels 3g is cited in text after 3h.

Line 568 and 685: Table S1 is mentioned, but I did not really find it in the article or the supplementary materials.

Line 637: this information is not contained within the reference 120.

Line 723: Despite, imitations in development: limitations

Table 3: Random lysines/ Breast cancer (near the reference 243),

Table 3 last row: Doxorubicine

Author Response

Comment 1: The authors have corrected the manuscript, which is now better organized, more systematically written, and the use of scientific terms is improved. They have also improved the quality of the Figures, which now better support the text. The only major remark I would have is that they mention Table S1, but it is not contained within the main text or in the Supplementary Files.

Response 1: We thank Reviewer 2 for his corrections and suggestions. We believe that the manuscript has been substantially improved. We are sorry for the mistake. We have uploaded this time the ‘Supplementary file’ where you can consult the new Table S1.

Please find below a short list of remarks:

Line 339: Prolactin receptor (PRLR): the abbreviation needs to be explained.

Corrected.

Lines 379-380: Figure panels 3g is cited in text after 3h.

Corrected.

Line 568 and 685: Table S1 is mentioned, but I did not really find it in the article or the supplementary materials.

We regret this shortcoming. We have incorporated the Supplementary into the new submission.

Line 637: this information is not contained within the reference 120.

Thank you for noticing. We have confused one of the references, which is now updated (reference 164). We have also checked that the rest of the citations are correct.

Line 723: Despite, imitations in development: limitations

Corrected.

Table 3: Random lysines/ Breast cancer (near the reference 243),

Thank you. Corrected.

Table 3 last row: Doxorubicine

Updated. Thank you.